# Evaluation of forecasts by a global data-driven weather model with and without probabilistic post-processing at Norwegian stations

John Bjørnar Bremnes, Thomas N. Nipen, and Ivar A. Seierstad

Norwegian Meteorological Institute

Box 43 Blindern, Oslo, Norway

**Correspondence:** John Bjørnar Bremnes (j.b.bremnes@met.no)

**Abstract.** During the last two years, tremendous progress in global data-driven weather models trained on numerical weather prediction (NWP) re-analysis data has been made. The most recent models trained on the ERA5 at $0.25°$ resolution demonstrate forecast quality on par with ECMWF's high-resolution model with respect to a wide selection of verification metrics. In this study, one of these models, the Pangu-Weather, is compared to several NWP models with and without probabilistic post-

processing for 2-meter temperature and 10-meter wind speed forecasting at 183 Norwegian SYNOP stations up to +60 hours ahead. The NWP models included are the ECMWF HRES, ECMWF ENS and the Harmonie-AROME ensemble model MEPS with 2.5 km spatial resolution. Results show that the performances of the global models are on the same level with Pangu-Weather being slightly better than the ECMWF models for temperature and slightly worse for wind speed. The MEPS model clearly provided the best forecasts for both parameters. The post-processing improved the forecast quality considerably for all

models, but to a larger extent for the coarse-resolution global models due to stronger systematic deficiencies in these. Apart from this, the main characteristics in the scores were more or less the same with and without post-processing. Our results thus confirm the conclusions from other studies that global data-driven models are promising for operational weather forecasting.

## 1 Introduction

Statistical machine learning methods have for decades been applied to calibrate or enhance weather forecasts based on output

from numerical weather prediction (NWP) models. Since NWP models already provide very good forecasts of most quantities, simple statistical models are often sufficient for extracting predictive information in NWP model output and making proba-bilistic forecasts. In constrast, recent advancements in deep learning methods have made it possible to develop more complex machine learning models based on the initial state only; first, in nowcasting of precipitation (Shi et al., 2015; Ravuri et al., 2021; Leinonen et al., 2023; Zhang et al., 2023) and more recently in medium-range weather forecasting of several parameters

for the entire atmosphere (Keisler, 2022; Pathak et al., 2022; Bi et al., 2023; Lam et al., 2022; Chen et al., 2023). The latter are trained on long archives of re-analysis data with coarse spatial resolution from the ERA5 (Hersbach et al., 2020, 2023) by learning one or more time-steps which in an auto-regressive manner are applied to generate complete forecasts several days ahead. The training process may take days or weeks on systems with multiple Graphical Processing Units (GPUs) or similar, but once trained predictions can be made in seconds or minutes on a single GPU. That is, only a tiny fraction of the cost of

NWP-based forecasting is required for forecast generation making these models very attractive for operational weather forecasting. The forecast accuracy of the models seems now to have reached the level of global NWP models for basic weather parameters, though, not all properties of these forecasts are yet well understood. Despite this, a few weather centres are now starting to routinely produce weather forecasts using global machine learning models initiated by NWP analyses in order to explore their full potential in more detail.

The aim of this study is to complement the verification results already reported by the machine learning model development teams and the study of Ben-Bouallegue et al. (2023), which evaluates the performance of the Pangu-Weather model (Bi et al., 2023). The Pangu-Weather model is based on a spatial 3D transformer network with input from selected surface variables and standard upper-air variables on 13 pressure levels only. Time steps of 1, 3, 6 and 24 hours are learned and combined to generate forecasts 10 days ahead. Models for each time step are trained on 192 NVIDIA V100 GPUs for 16 days using 39 years of hourly ERA5 data. In this article, the attention is paid to temperature and wind speed forecasts up to 60 hours ahead generated by the Pangu-Weather model at a set of Norwegian SYNOP measurement stations. The forecasts are compared against operational forecasts from the ECMWF IFS models HRES and ENS and the Harmonie-AROME model MEPS (Frogner et al., 2019; Andrae et al., 2020) for northern Europe at 2.5 km spatial resolution. In order to make our conclusions more robust, forecasts from all models are separately post-processed using the Bernstein quantile network method (Bremnes, 2020; Schulz and Lerch, 2022). The post-processing is capable of reducing obvious systematic deficiencies which otherwise may influence the inter-comparison.

## 2 Data

In order to compare the forecast models, a data set for 2m temperature and 10m wind speed was collected and organised for 183 Norwegian SYNOP stations for the years 2021 and 2022. The locations of the stations are shown in Fig. 1 along with the topography of Norway. The majority of the stations are situated along the coastline and fjords and in the valleys. Only a minority of the stations are at higher elevations. Observations with a six hourly temporal resolution were extracted from the Norwegian Meteorological Institute's observation database. Only stations with more than 75% data availability during the two-year period were included. All observations have undergone an extensive automatic quality control and to some degree a manual assessment.

Forecasts from the Pangu-Weather (PANGU) model were generated using software tools[1] provided by ECMWF with initial states from the ERA5 re-analysis with 0.25° spatial resolution. Further, three operational NWP models were included: ECMWF HRES available on a latitude-longitude grid of 0.1° resolution, ECMWF ENS with 51 ensemble members at 0.2° resolution, and the Harmonie-AROME model MEPS (Frogner et al., 2019; Andrae et al., 2020) for northern Europe with 15 lagged ensemble members in a 3-hour time window at 2.5 km resolution. In addition, the control members of the ENS and MEPS ensembles were included in the experiments as separate deterministic models and denoted by ENS0 and MEPS0, respectively.

---

[1]https://github.com/ecmwf-lab/ai-models-panguweather

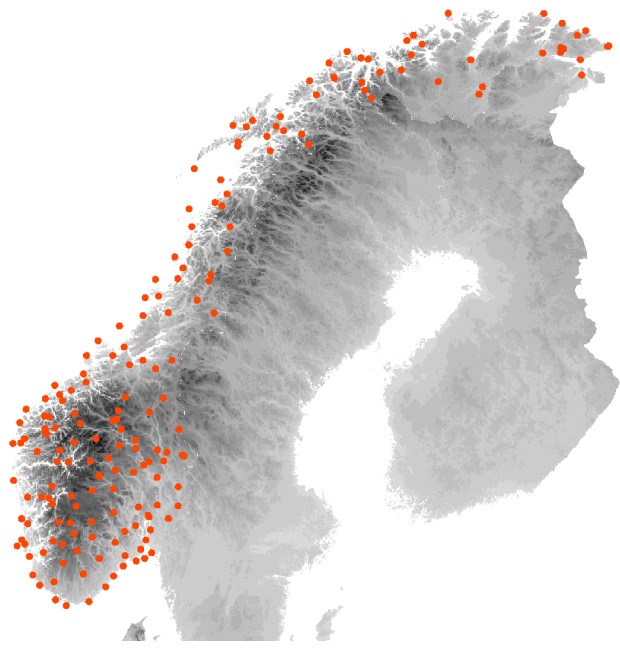

**Figure 1.** Location of the 183 Norwegian stations with elevation in the background. The darkest gray shade corresponds to an altitude of 2400 meters.

All forecasts were initiated at 00 UTC with lead times +6h, +12h, ..., +60h and bi-linearly interpolated to the locations of the stations.

Data for the year 2021 was dedicated to training post-processing models and further split in two sets: the first 25 days in each month were used for training the models and the remaining days for selecting the best models while training as described in section 4. In total there were 512,840 cases available for training, 107,515 for validation and 602,909 independent cases for comparing all models in the end. Data for the exact same times were used for all models both for training and final evaluation.

## 3 Methods

In this section the probabilistic post-processing method and the verification metrics are described.

### 3.1 Bernstein Quantile Networks

There is a wide selection of post-processing methods available for making probabilistic forecasts based on either deterministic or ensemble input, e.g. Vannitsem et al. (2021). In this study, the Bernstein Quantile Networks (BQN) (Bremnes, 2020) is chosen due to its adaptability to any continuous target variable. The key properties of the BQN are: first, the forecast distribution is specified by its quantile function and assumed to be a Bernstein polynomial. Second, the distribution is linked to the input

variables by means of a neural network taking any relevant set of variables as input and outputs the coefficients of the Bernstein polynomial. The parameters of the neural network are then in practice the distribution parameters and can be estimated by optimising the quantile score (see below) over a pre-defined set of quantile levels. The degree of the Bernstein polynomial determines the flexibility of the distribution. By choosing a sufficiently high degree, any shape of the distribution can in principle be allowed for.

Let $x$ be the vector of input variables and $\tau \in [0,1]$ the quantile level. The quantile function is then defined as

$$Q(\tau|x) = \sum_{j=0}^{d} \alpha_j(x)\binom{d}{j}\tau^j(1-\tau)^{d-j} \tag{1}$$

where $d$ is the degree of the Bernstein polynomial with coefficients $\alpha_j(x)$ that are functions of the input variables $x$ or, more precisely, a neural network with input $x$. The subsequent terms are the Bernstein basis polynomials of degree $d$ which only depend on the quantile level $\tau$. To ensure that the quantile function is valid, that is, non-decreasing with increasing $\tau$, it is sufficient to constrain the $\alpha_j(x), j = 0, 1, ..., d$ to be in non-decreasing order for any $x$ which can be obtained by a re-parameterisation of the coefficients, see Reich et al. (2011); Schulz and Lerch (2022) for further details.

Given a training data set of input variable and observation pairs $(x_i, y_i)$, $i = 1, ..., n$ and a set of quantile levels $\tau_1, ..., \tau_T$ the parameters of the BQN model can be estimated by minimising

$$\sum_{i=1}^{n}\sum_{t=1}^{T}\rho_{\tau_t}(y_i - Q(\tau_t|x_i)) \tag{2}$$

with respect to the underlying parameters of the $\alpha_j(x)$, that is, the weights and biases of the neural network. The quantile loss function $\rho_\tau$ is defined by

$$\rho_\tau(u) = \begin{cases} u(\tau - 1) & u < 0 \\ u\tau & u \geq 0. \end{cases} \tag{3}$$

The optimisation problem in Eq. 2 is solved numerically by stochastic gradient descent, see section 4.1 for further details.

## 3.2 Forecast and verification metrics

In our experiments the accuracy and properties of both deterministic and probabilistic/ensemble forecasts are assessed. The metrics applied for deterministic forecasts are listed in Table 1 along with a description of how they are calculated. These include basic scores like the mean absolute error (MAE), mean error (ME) and standard deviation of error (SDE), but also measures of forecast and observed variability as well as extremes. For deterministic forecast verification, ensemble and probabilistic forecasts are reduced to their medians or 50 percentiles which are optimal with respect to absolute error.

The continuous ranked probability score (CRPS) (Matheson and Winkler, 1976; Gneiting and Raftery, 2007) which is a proper scoring function is chosen as the summarising measure for ensemble and probabilistic forecasts. Given a single ensemble forecast of $m$ members, $e_1, e_2, ..., e_m$, and a corresponding observation $y$ the CRPS is defined by

$$CRPS = \frac{1}{m}\sum_{i=1}^{m}|e_i - y| + \frac{1}{2m^2}\sum_{i=1}^{m}\sum_{j=1}^{m}|e_i - e_j| \tag{4}$$

| Metric | Abbreviation | Description |
|---|---|---|
| mean absolute error | MAE | mean absolute difference of forecasts and observations |
| mean error (bias) | ME | mean difference of forecasts and observations |
| standard deviation of error | SDE | standard deviation of error computed for each station and then averaged |
| standard deviation ratio | SDR | standard deviation of forecasts divided by standard deviation of observations calculated for each station and then averaged |
| deviation in maxima | | difference between maximum forecast value and maximum observed value calculated for each station and then averaged |
| deviation in minima | | difference between minimum forecast value and minimum observed value calculated for each station and then averaged |
| maxima ratio | | maximum forecast divided by maximum observation calculated for each station and then averaged |

**Table 1.** List of verification metrics for deterministic forecasts.

The CRPS is negatively oriented, that is, lower CRPSs are preferred. In case of only one member the CRPS reduces to the absolute error. For several forecasts the CRPS is just computed for each of the given forecasts and then averaged. In this study, the definition in Eq. 4 is also applied for the quantile forecasts generated by the BQN method. Further, it could be noted that in this study, CRPSs for ensembles and sets of quantiles of different sizes are compared, which may to a minor extent favour the largest sized models (Ferro et al., 2008).

## 4 Experiments and results

In this section details on the post-processing models are first provided followed by the presentation and discussion of the results for the temperature and wind speed data sets.

### 4.1 Description of post-processing models

The BQN method is applied to make probabilistic forecasts of temperature and wind speed at the location of the stations using forecasts from Pangu-Weather and the NWP models as input, that is, separate BQN models for each parameter and input model. Since both deterministic (PANGU, HRES, MEPS0 and ENS0) and ensemble models (MEPS and ENS) are included in the study, different input variables to the BQN models were selected as listed in Table 2. From the input models only the relevant forecast variable was chosen, though for wind both its magnitude, zonal and meridional components were included. For ensembles, the ensemble mean and standard deviation were used. In addition to the forecast model variables, a set of static predictors was added. Simple trigonometric functions of the day of year was applied to account for possible seasonal variations. Lead time was included for the simplicity of having one BQN model for all lead times. Variation between stations was allowed for by including the station id as an embedding in the network before it was merged with the other input variables.

| Input variable | Temperature | | Wind speed | |
|---|---|---|---|---|
| | deterministic | ensemble | deterministic | ensemble |
| station id | × | × | × | × |
| lead time | × | × | × | × |
| cosine day of year: cos(2πd/365) | × | × | × | × |
| sinus day of year: sin(2πd/365) | × | × | × | × |
| temperature | × | | | |
| ensemble mean of temperature | | × | | |
| ensemble standard deviation of temperature | | × | | |
| wind (magnitude, zonal and meridional) | | | × | |
| ensemble mean of wind (magnitude, zonal and meridional) | | | × | × |
| ensemble standard deviation of wind speed | | | × | × |

**Table 2.** List of input variables for BQN temperature and wind speed models with deterministic and ensemble inputs. The columns for deterministic input refer to the PANGU, HRES, MEPS0 and ENS0 models, while ensemble columns refer to MEPS and ENS. Wind is represented by three variables, its magnitude and zonal and meridional components.

In the embedding each station id was represented by eight trainable parameters in order to capture spatial variability. With 183 stations this amounted to $183 \times 8 = 1464$ parameters in total for the embedding.

The definition of the neural network was primarily based on the studies in Bremnes (2020); Schulz and Lerch (2022) without further hyper-parameter tuning on the data in this study. The number of fully-connected/dense layers in the network was set to two with elu activation functions and a softplus transformation at the end to constrain the Bernstein coefficients to be in increasing order. The degree of the Bernstein polynomials was set to 12 which allowed for highly flexible forecast distributions. Other hyper-parameter choices and model details were as follows

- batch size of 128 with random shuffling of the data between the epochs

- continuous input variables standardised by subtracting means and dividing by standard deviations

- early stopping with initial learning rate of 0.001 with a reduction by a factor of 10 if no improvement on the validation set was not obtained in 10 epochs. The maximum number of epochs was set to 200 and minimum learning rate to $5 \cdot 10^{-6}$.

- ADAM optimiser with default parameters

- quantile loss function averaged over quantile levels 0.025, 0.050, ..., 0.975

- predictions for the 0.00, 0.01, ..., 1.00 quantiles

Three network models were fitted with different numbers of units in the two dense layers: (64, 32), (32, 32) and (32, 16). In addition, each of these was trained three times with random initial parameters. All verification statistics on the test data set were averaged over these nine models to make the results more robust to model specification and random variations.

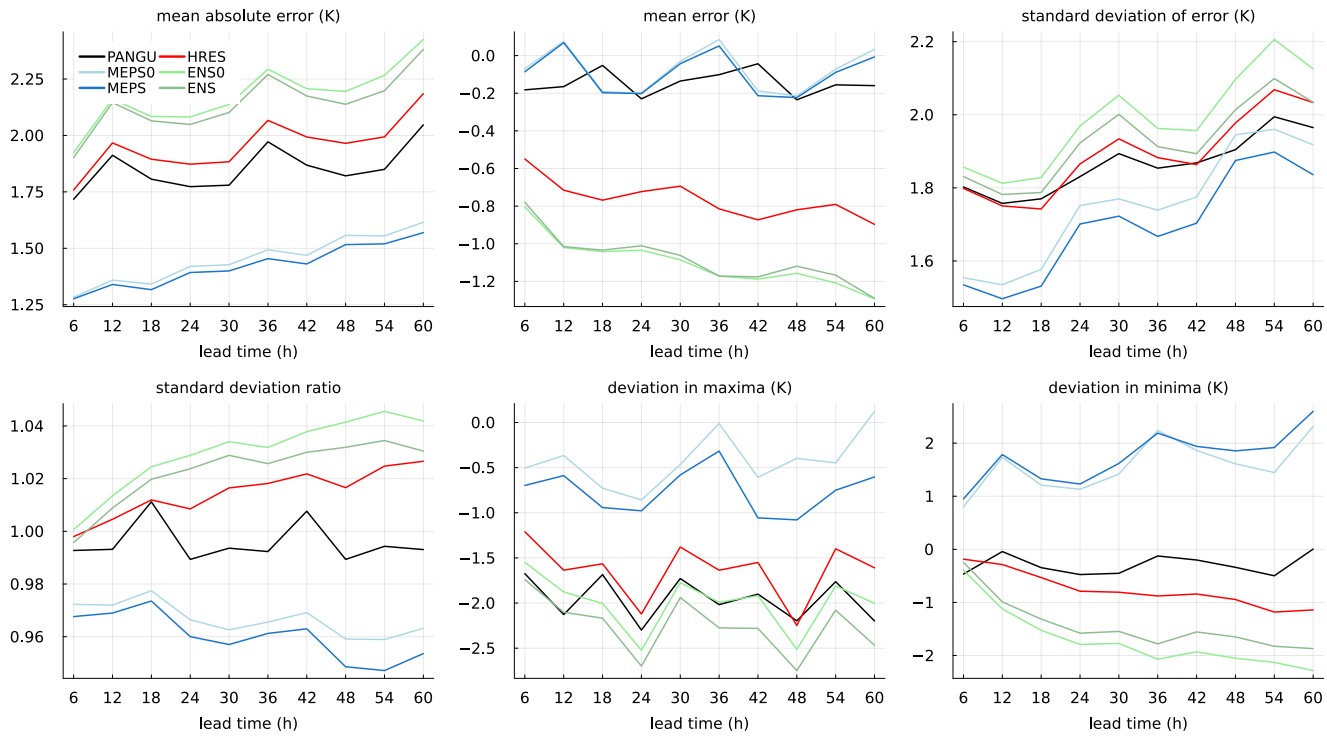

**Figure 2.** Deterministic forecast and verification metrics for temperature at 2 meter as a function of lead time for the forecast models without post-processing.

## 4.2 Temperature at 2 meter

In Fig. 2 deterministic verification statistics for the NWP models and Pangu-Weather without any post-processing are shown as a function of lead time. With respect to MAE, MEPS (ensemble median) and MEPS0 (control member) are considerably better than the rest. The MAE for MEPS is slightly lower than for MEPS0 as expected due to the fact that the median is optimal with respect to absolute error. Further, PANGU scores better than both HRES and the ENS despite being trained on coarse resolution ERA5 data. On the other hand, it may have an advantage of being initiated with re-analysis data instead of an operational NWP analysis (Ben-Bouallegue et al., 2023). Concerning the mean error (bias) the MEPS models and PANGU have scores close to zero. The negative mean errors of HRES and ENSs are possibly due to elevation differences and a cold-bias along the coast during winter (not shown). It is, however, surprising that this is not the case for PANGU. In the SDE, the mean errors are subtracted for each station before squaring the error, thus it is invariant to possible biases. The pattern is the same as for MAE, but the differences between the models are less in percentage, in particular for the longer lead times. The SDEs for PANGU and HRES are also more similar. Forecast variation is summarised by the standard deviation ratio. HRES, ENS0 and ENS have slightly larger variation than the observations. One possible explanation may be that with a coarse model resolution, lakes and sea are not well resolved such that the sites on average are less influenced by sea/lake temperatures than in the reality. The

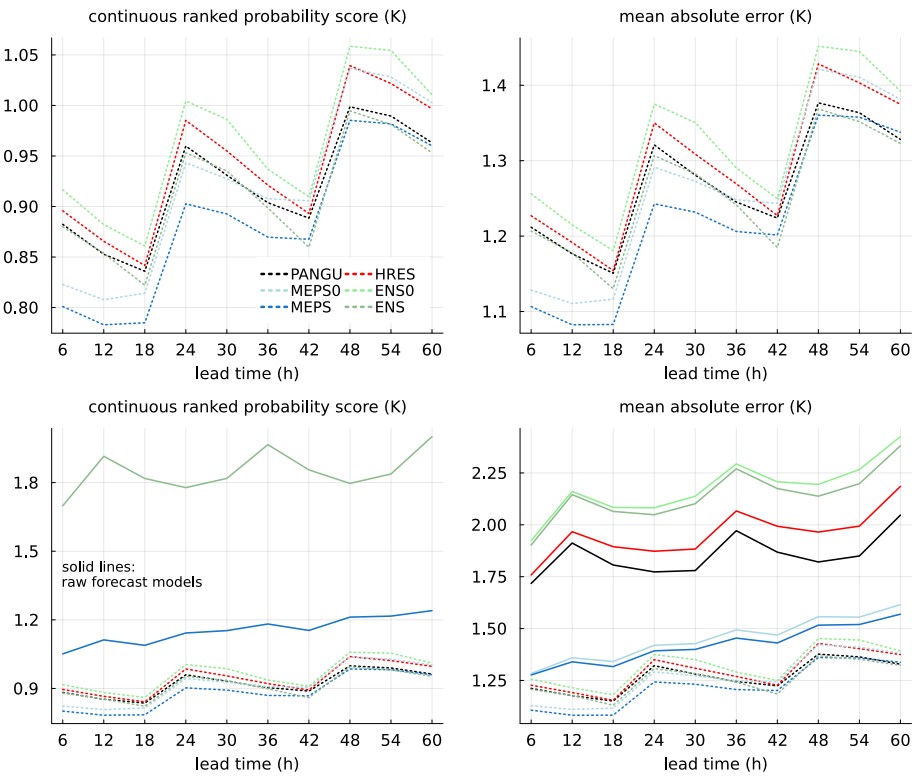

**Figure 3.** Continuous ranked probability score and mean absolute error for temperature at 2 meter as a function of lead time. In upper row scores for BQN calibrated forecasts only (dotted lines). In lower row raw forecast models (solid lines) are added for comparison.

average model elevation at the sites for these models are also somewhat higher than the actual elevation, although it is unknown whether this can have an impact on forecast variability. However, PANGU, which has about the same model orography as the ECMWF models, has about the same variability as the observations. Both MEPS and MEPS0 forecasts have less standard deviation than observed. Concerning temperature maxima, all models are on average over the stations too cold, in particular PANGU and the ECMWF models. On annual minimum temperature the MEPS models are too warm, while the rest are too cold.

In Fig. 3 the CRPS and MAE of the medians/50 percentiles are shown for the models after post-processing as well as for the uncalibrated forecasts. From the the lower panels it can be noticed that post-processing improves the models considerably. For the ECMWF models and PANGU the reductions are up to about 50%, especially for ENS and ENS0. For MEPS and MEPS0 the improvement are less, but still highly significant. Concerning the calibrated models there is a distinct difference in skill between models with ensemble input, MEPS and ENS, and their deterministic variants using only the control members (MEPS0 and ENS0) confirming the usefulness of ensemble systems. After calibration the difference between the models become less, in particular for longer lead times. This is due to stronger systematic errors in the models with coarser resolution which in

|        | PANGU | HRES | ENS  | MEPS | ENS0 | MEPS0 |
| ------ | ----- | ---- | ---- | ---- | ---- | ----- |
| PANGU  |       | 33.6 | 28.4 | 14.4 | 42.5 | 20.3  |
| HRES   | 27.9  |      | 23.7 | 9.3  | 44.3 | 17.6  |
| ENS    | 39.1  | 46.1 |      | 16.1 | 91.5 | 26.1  |
| MEPS   | 45.5  | 49.2 | 40.1 |      | 53.6 | 72.2  |
| ENS0   | 20.9  | 23.1 | 1.6  | 6.3  |      | 11.9  |
| MEPS0  | 28.9  | 32.2 | 24.8 | 3.6  | 36.3 |       |

**Table 3.** Proportion (%) of site and lead time combinations (183×10 in total) where the model in a given row is statistically significantly better than those in the columns with respect to CRPS. The results are for post-processed 2m temperature forecasts.

general are easier to improve upon by post-processing methods. However, the high-resolution MEPS model still have clearly better scores than the global models, at least up to 42 hours ahead. It should also be mentioned that the ECMWF models have longer data-assimilation window than the MEPS model making their effective lead time shorter.

To assess whether the differences in CRPS for the post-processed models are statistically significant, one-sided pairwise Diebold and Mariano (1995) hypothesis tests are applied separately for each site and lead time and the number of significant outcomes are counted, see Appendix A for more details. The testing procedure is carried out individually for all pairs of models and the results are summarised in Table 3. As expected, PANGU and HRES have roughly equal fractions of sites and lead times where either of them is best. Further, the global models are for very few combinations (up to 16.1%) better than MEPS and these are likely for the longer lead times where the scores are most similar. The most significant results are for ensemble versus control member models; ENS is in 91.5% of cases significantly better than its control member (ENS0), while MEPS is better than MEPS0 in 72.2% of the combinations. Overall one might have expected that the outcome of the tests would result in proportions closer to 0% or 100%, but it should be kept in mind that there are only up to 365 forecast cases in each pairwise test. With a nominal significance level of 0.05 the differences are often too small to be of statistical significance with the current data size. An alternative would be to apply the testing procedure separately for each lead time. For the shorter lead times it is reasonable to anticipate stronger significance results for MEPS for instance.

## 4.3 Wind speed at 10m

Deterministic verification metrics for the 10m wind speed forecasts without post-processing are presented in Fig. 4. As expected, MEPS and MEPS0 are clearly better than HRES, ENS, ENS0 and PANGU with respect to MAE. Unlike for temperature, the ECMWF models have slightly better scores than PANGU for all lead times. The differences in MAE between the ECMWF models and MEPS models are partly due to the relative large systematic underestimation of wind speed for the ECMWF models as shown in the mean error panel. A negative bias can be noticed for all models, but in particular for HRES, ENS, ENS0 and PANGU. The standard deviation of error reveals the same pattern as for MAE with the largest differences for short lead times. That is, with decreasing predictability the differences become less clear. For longer lead times high-resolution

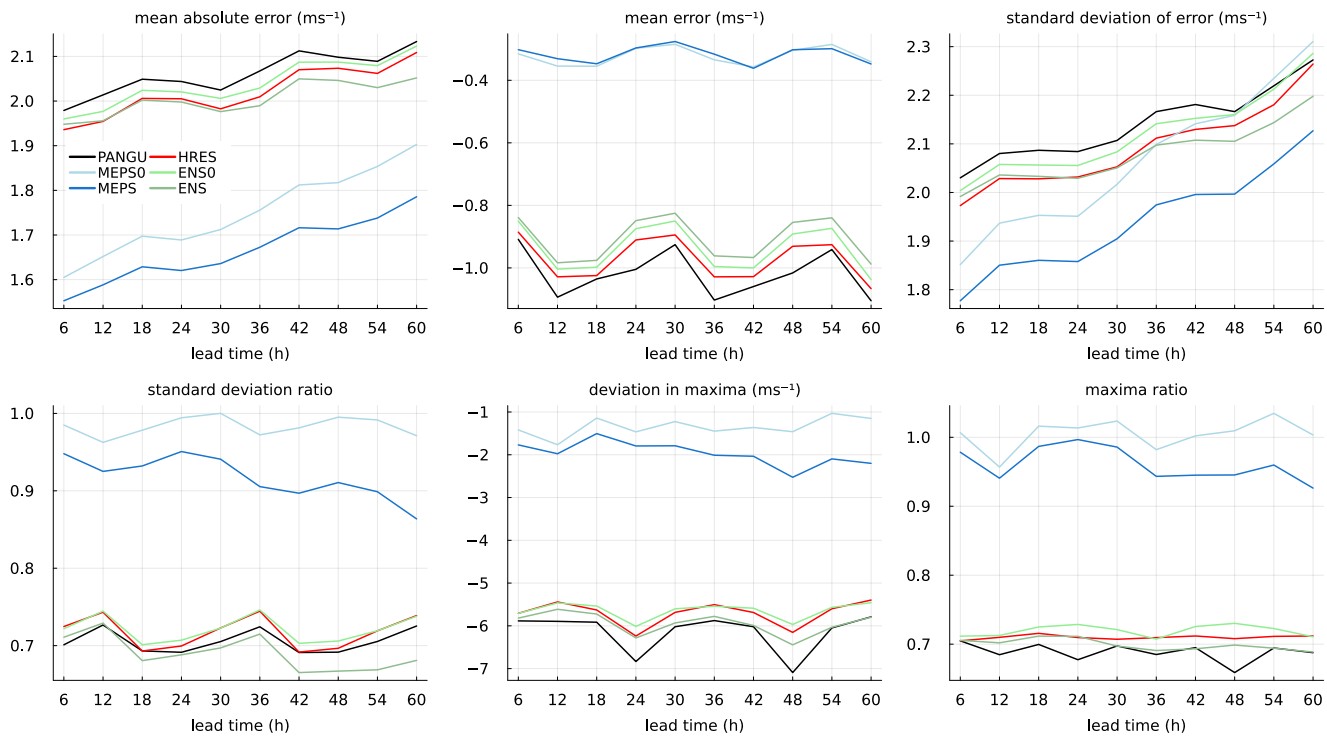

**Figure 4.** Forecast and verification metrics for wind speed at 10 meter as a function of lead time for the forecast models without post-processing.

NWP models are more prone to displacement errors in time and space than models with smoother forecast fields. This may have an impact in verification against point measurements as here. The forecast variability of HRES, ENS, ENS0 and PANGU are only about 70% of the observed variability as quantified by the standard deviation ratio. These models also severely underestimate extreme wind speeds. Averaged over the stations the annual forecast wind speed maxima are about 6 ms$^{-1}$ weaker than the observed maxima. The panel showing the maxima ratio also confirms this. For MEPS the model climatology is much closer to the observed. For the control member (MEPS0), the standard deviation and maxima ratios are close to one, while there is an underestimation of about 1.5 ms$^{-1}$ of the annual extremes. The MEPS (ensemble median) has less variability than MEPS0 as expected by construction.

CRPS and MAE for the post-processed probabilistic forecasts and the raw models are shown in Fig. 5. In the lower panels the benefits of post-processing is evident. For example, for ENS the CRPS is reduced by up to about 50% and for MEPS about 35%. This can partly be attributed to BQN models being capable of learning local systematic deviations. For wind speed there may be strong fine-scale variability in the proximity of most stations that NWP models are not able to represent. Further, models based on ensemble input are consistently better than deterministic ones, thus, again confirming the usefulness of ensemble

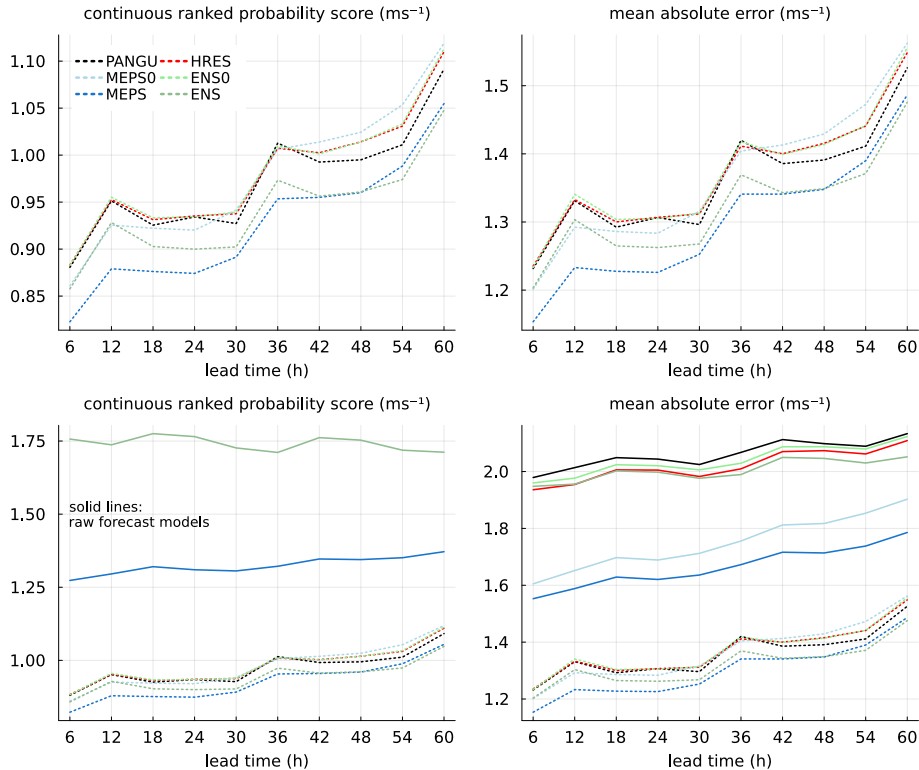

**Figure 5.** Continuous ranked probability score and mean absolute error for wind speed at 10 meter as a function of lead time. In upper row scores for BQN calibrated forecasts only (dotted lines). Lower row also including raw forecast models (solid lines).

forecast systems. For the longer lead times Pangu-Weather seems to benefit more of post-processing than the ECMWF models which could be due to smoother forecast fields, but this needs further investigations.

As for temperature, testing statistical significance of differences in CRPS was performed for the post-processed wind speed forecasts in a pairwise manner. The proportions of site and lead time combinations where each model is better than the remain-
200 ing models are given in Table 4. The difference between PANGU and HRES is clearly negligible. MEPS is significantly better than the global models for roughly half of the 183×10 site and lead time combinations which is mostly due to the differences for the shorter lead times.

## 5 Conclusions

In this study we have compared the forecast accuracy of the Pangu-Weather model with global NWP models from ECMWF and
205 the high-resolution limited area model MEPS by verifying against Norwegian SYNOP measurements. In addition, probabilistic post-processed forecasts from all ensemble and deterministic models have been evaluated. The latter is especially useful for better showing the full potential in each of the models as it is capable of removing systematic deviations. Besides, in many

|        | PANGU | HRES | ENS  | MEPS | ENS0 | MEPS0 |
|--------|-------|------|------|------|------|-------|
| PANGU  |       | 27.5 | 15.2 | 7.0  | 32.8 | 20.6  |
| HRES   | 28.9  |      | 18.1 | 9.2  | 36.6 | 24.9  |
| ENS    | 50.4  | 54.5 |      | 19.3 | 91.6 | 40.5  |
| MEPS   | 54.8  | 52.1 | 42.3 |      | 56.4 | 88.2  |
| ENS0   | 27.7  | 31.0 | 2.8  | 8.3  |      | 23.6  |
| MEPS0  | 23.9  | 24.4 | 15.9 | 0.5  | 27.5 |       |

**Table 4.** Proportion (%) of site and lead time combinations (183×10 in total) where the model in a given row is statistically significantly better than those in the columns with respect to CRPS. The results are for post-processed 10m wind speed forecasts.

forecast products for end users, post-processing is frequently applied making our comparisons even more relevant. In this sense, our study have complemented the assessments of Bi et al. (2023); Ben-Bouallegue et al. (2023).

The main finding of our work is that Pangu-Weather is slightly better than the ECMWF models for temperature and vice versa for wind speed. For temperature the differences are mostly due to less bias in Pangu-Weather, while for wind speed differences in bias are less obvious. The global models are, however, considerably less skillful than the high-resolution MEPS model. After post-processing the differences decrease significantly due to larger systematic errors in the coarse-scale Pangu-Weather and ECMWF models which post-processing methods effectively can remove. For the longer lead times the summarising scores are highly similar. Our study also clearly shows the advantage of NWP ensemble forecasting.

There is certainly scope for further work. A more in-depth evaluation at station and seasonal levels would be needed to discover more of the characteristics of Pangu-Weather and global data-driven machine learning models in general. Another and much debated topic, which we have not looked into, is the amount of space-time smoothing with increasing lead time. Regarding post-processing methods, it would be useful to set up experiments with more predictor variables, or even different methods, and study their impacts. Further, with post-processing it would be interesting to evaluate predictions beyond point measurements; for example, gridded products or space-time functionals like the maximum wind speed over a larger area during a 12-hour period, say, would be instructive. If other high-impact variables like precipitation become available, then these would be of interest for an inter-comparison. The same would apply to forecast products tailored to specific end users, for example within the energy sector. Since the Pangu-Weather model was published further improvements in global machine learning models have been made Lam et al. (2022); Chen et al. (2023). A follow-up study with post-processing of several machine learning models would be interesting as well as ensembles generated by purely data-driven machine learning models. Finally, there is ongoing work on training global graph-based machine learning models with kilometre scale resolution over regions of special interest and coarser resolution elsewhere. These models make use of long archives of high-resolution limited area reanalysis data in combination with global reanalyses like ERA5. Comparing these with limited area NWP models like MEPS will be of great interest.

*Code and data availability.* The methods and analyses are implemented in Julia based on the Flux package for machine learning (Innes et al., 2018; Innes, 2018). Source codes are made available on GitHub at https://github.com/jbbremnes/pangu-asr. The data (about 3 GB) can be downloaded from https://zenodo.org/records/10210204 (DOI 10.5281/zenodo.10210203).

## Appendix A: Testing statistical significance

Testing statistical significance is challenging due to complex dependencies in space and time and there are several ways forward. Hence, it is important to be aware that the outcome would depend on how such tests are set up and the purpose. We have chosen to focus on site and lead time specific performance by applying the Diebold and Mariano (1995) test separately for each site and lead time and using the Benjamini and Hochberg (1995) procedure to control the false discovery rate at the given nominal level. Before the testing procedure is applied, each forecast model is averaged over its 3×3 variants of initial states

and network configurations, see section 4.1. The testing procedure is here only applied to the post-processed forecasts. The proportion of the 183×10 site and lead time combinations where a given model is significantly better than the alternative model at the 0.05 nominal level is reported. This is the same procedure that was applied by Schulz and Lerch (2022) and suggested by Wilks (2016). We refer to the references for further mathematical details.

*Author contributions.* JBB prepared the NWP and observational data, generated the post-processed forecasts, computed the verification

statistics and prepared the manuscript. TN generated the Pangu-Weather forecasts. TN and IS contributed to the analysis of the results and in finalising the article.

*Competing interests.* The authors declare that they have no conflict of interest.

*Acknowledgements.* We would like to acknowledge Huawei Cloud for making the trained Pangu-Weather model available, ECMWF for the software to generate Pangu-Weather forecasts and Jørn Kristiansen and two anonymous reviewers for their feedback on the manuscript.

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
