# Peer review of "Evaluation of forecasts by a global data-driven weather model with and without probabilistic post-processing at Norwegian stations"

_EGUsphere, 2023_

## Author Response (AR1)

**Author's response**

egusphere-2023-2838

Title: Evaluation of forecasts by a global data-driven weather model with and without probabilistic post-processing at Norwegian stations
Author(s): John Bjørnar Bremnes et al.
MS No.: egusphere-2023-2838
MS type: Research article

Online discussion:
https://egusphere.copernicus.org/preprints/2023/egusphere-2023-2838/#discussion

Author's comments online are in blue and additional comments for the revision of the manuscript in red.

**Referee #1 (RC1)**

Over the past two years, there has been rapid and unprecedented progress in data-driven, AI-based models for weather prediction. The paper evaluates the forecast quality of Pangu-Weather, a state-of-the-art AI weather model, and two physics-based NWP models: the ECMWF ensemble, and MEPS, a high-resolution limited area model. The forecasts are compared based on temperature and wind speed data from observation stations in Norway. Overall, the authors find that the unprocessed forecasts of the MEPS model are superior to those of the ECMWF ensemble and the Pangu-Weather model (which perform similarly). After post-processing, the relative differences in terms of the evaluation metrics are much smaller, with slight advantages for the post-processed MEPS forecasts.

In my view, the paper is timely and addresses interesting and important research questions given the recent rise in data-driven weather forecasting. The two main contributions are that

- the paper provides (potentially the first, but at least one of the first) comparisons of AI-based and physics-based weather forecasting models based on station data (rather than the commonly used comparisons based on gridded ERA5 data);
- the paper assesses and quantifies the effect of post-processing on forecasts from AI-based weather models. The paper is well written; the findings and conclusions are presented clearly and are supported by the presented results. Some minor comments, questions and suggestions are summarized below.

General comments:

1. To me, the most interesting contribution was on the effect of post-processing on the AI-based vs. physics-based weather models. Though not entirely unexpected, I found it interesting to see that after post-processing, the differences between the ECMWF and

Pangu-Weather forecasts seems to be minimal. This contribution of the paper might be strengthened by highlighting it more, for example in the 'Conclusions' section.

2. Regarding the first comment from above, it might have been interesting to add some notion of 'significance' or uncertainty to the observed score differences, in particular after post-processing. I would expect that (potentially except for parts of the MEPS scores) the differences likely are not significant.

3. In terms of post-processing methods, only BQN is applied. Even though I assume that this is not the case, it might be interesting to discuss whether there are any reasons to assume whether different effects of post-processing could potentially be expected for other post-processing approaches.

4. The BQN post-processed forecasts only use weather variables directly related to the target variable. However, it has been demonstrated in various papers (on physics-based weather models) that neural network-based post-processing methods such as BQN can benefit substantially from including additional predictors. Would you expect similar improvements when utilizing additional predictors from AI-based weather models?

5. In the presentation of the results in Figures 2 and 3 (and 4 and 5), I found it somewhat confusing that the line types do not consistently refer to raw and post-processed forecasts. Personally, I would have found it more instructive, if (for example), scores for deterministic forecasts are always drawn in solid lines, and forecasts for probabilistic scores in dashed lines (or vice versa).

**Author's response** (AC1)

We would like to thank the reviewer for the feedback on the article. Below is our initial response:

1) Yes, we will add this to the concluding section.

The following is added to the concluding section: "After post-processing the differences decrease significantly due to larger systematic errors in the coarse-scale Pangu-Weather and ECMWF models which post-processing methods effectively can remove. For the longer lead times the summarising scores are highly similar."

2) Testing statistical significance is challenging due to complex dependencies in space and time and there are several ways forward. Hence, it is important to be aware that the outcome would depend on how such tests are set up. However, we agree that it would be useful to have some idea of the significance of the differences. We have chosen to focus on site and lead time specific performance by applying the Diebold-Mariano (1995) test separately for each site and lead time and using the Benjamini-Hochberg (1995) procedure to control the false discovery rate at the given level. In the tables below, performance in terms of CRPS for the post-processed forecasts is considered. The reported figures are the percentages of the 183×10 site and lead time combinations where the model in a given row is significantly better than the models in the columns at the 0.05 nominal level.

```
Temperature 2m
         | pangu    hres    ens    meps    ens0   meps0
---------+------------------------------------------------
pangu  |   0.0    33.6    28.4    14.4    42.5    20.3
hres   |  27.9     0.0    23.7     9.3    44.3    17.6
ens    |  39.1    46.1     0.0    16.1    91.5    26.1
meps   |  45.5    49.2    40.1     0.0    53.6    72.2
ens0   |  20.9    23.1     1.6     6.3     0.0    11.9
meps0  |  28.9    32.2    24.8     3.6    36.3     0.0

Wind speed 10m
         | pangu    hres    ens    meps    ens0   meps0
---------+------------------------------------------------
pangu  |   0.0    27.5    15.2     7.0    32.8    20.6
hres   |  28.9     0.0    18.1     9.2    36.6    24.9
ens    |  50.4    54.5     0.0    19.3    91.6    40.5
meps   |  54.8    52.1    42.3     0.0    56.4    88.2
ens0   |  27.7    31.0     2.8     8.3     0.0    23.6
meps0  |  23.9    24.4    15.9     0.5    27.5     0.0
```

We will include more details on the testing procedure and update the text accordingly in the revised version of the manuscript. We will also consider making a test on a more aggregated level. On the combined site and lead time level there are only up to 365 forecasts in each DM test, while in total there are 602,909 forecasts in the dataset for evaluation.

The table and following text is added to sections 4.2 and 4.3, respectively:
 "To assess whether the differences in CRPS for the post-processed models are statistically significant, one-sided pairwise \cite{diebold1995} hypothesis tests are applied separately for each site and lead time and the number of significant outcomes are counted, see Appendix A for more details. The testing procedure is carried out individually for all pairs of models and the results are summarised in Table \ref{tab:t2test}. As expected, PANGU and HRES have roughly equal fractions of sites and lead times where either of them is best. Further, the global models are for very few combinations (up to 16.1\%) better than MEPS and these are likely for the longer lead times where the scores are most similar. The most significant results are for ensemble versus control member models; ENS is in 91.5\% of cases significantly better than its control member (ENS0), while MEPS is better than MEPS0 in 72.2\% of the combinations. Overall one might have expected that the outcome of the tests would result in proportions closer to 0\% or 100\%, but it should be kept in mind that there are only up to 365 forecast cases in each pairwise test. With a nominal significance level of 0.05 the differences are often too small to be of statistical significance with the current data size. An alternative would be to repeat the testing procedure separately for each lead time. For the shorter lead times it is reasonable to anticipate stronger significance results for MEPS for instance."

"As for temperature, testing statistical significance of differences in CRPS was performed for the post-processed wind speed forecasts in a pairwise manner. The proportions of site and lead time combinations where each model is better than the remaining models are given in Table \ref{tab:s10test}. The difference between PANGU and HRES is clearly negligible. MEPS is significantly better than the global models for roughly half of the 183×10 site and lead time combinations which is mostly due to the differences for the shorter lead times."

An appendix describing the testing procedure is also added: "Testing statistical significance is challenging due to complex dependencies in space and time and there are several ways forward. Hence, it is important to be aware that the outcome would depend on how such tests are set up and the purpose. We have chosen to focus on site and lead time specific performance by applying the \cite{diebold1995} test separately for each site and lead time and using the \cite{benjamini1995} procedure to control the false discovery rate at the given nominal level. Before the testing procedure is applied, each forecast model is averaged over its 3×3 variants of initial states and network configurations, see section 4.1. The testing procedure is here only applied to the post-processed forecasts. The proportion of the 183×10 site and lead time combinations where a given model is significantly better than the alternative model at the 0.05 nominal level is reported. This is the same procedure that was applied by \cite{schulz2022} and suggested by \cite{wilks2016}. We refer to the references for further mathematical details."

3) Based on the research literature, it is reasonable to expect that other post-processing methods could provide forecasts of about the same skill. The choice of input data could, however, make a noticeable difference. In this study, only the most relevant variable interpolated to the site at the given (lead) time is used as input. As mentioned in 4) more input variables could very likely improve the performances. The same goes for including forecasts in a neighborhood around the given point in space and time. Further, the latter may have different effects on the various NWP models. For example, it could be that the high-resolution MEPS model (2.5 km) could benefit more from this than the NWP models with coarse resolutions, since the former are more prone to phase shifts in time and space with increasing lead time, in particular for wind speed.

4) Yes, including more variables to the input generally improves the scores. We do not see any reason why this should not be the case for data-driven AI models as well.

5) We will consider whether there is a better alternative.

We tend to agree with the reviewer that it would be better to have consistent line styles throughout and have therefore made changes in figures 3 and 5. In these, the calibrated forecasts are now represented by dotted lines and the raw forecast models by solid lines as in figures 2 and 4.

**Referee #2 (RC2)**

This manuscript is very well written. I like its length and clarity, and I believe it adds further evidence that ML/AI will revolutionize weather prediction. I have one minor technical comment (below); otherwise, I think the manuscript is well done and ready for full publication.

Page 3: Add a (ME) after mean error (second line from bottom) for consistency with other metric definitions and Table 1.

**Author's response** (AC2)

Will be done.

**Referee #2 (RC3)**

You conclude that the models, including Pangu-Weather, are considerably less skillful than the high-resolution MEPS model.

a) Does this make a strong case for a high-resolution/limited area re-analysis dataset?

b) Along those same lines, given a high resolution re-analysis dataset to train on, would you expect that ML/AI models will be equally competitive or better than traditional high-resolution NWP models and ensembles, and that the ML/AI approach will be equally skillful at predicting higher resolution, higher-impact meteorological phenomena? Or would high-impact meteorological prediction be better served by additional post-processing of the current resolution of Pangu-Weather and other global ML/AI models?

**Author's response** (AC3)

Concerning a), one way forward is to make global ML models with higher resolution over the area of interest by for example using a graph-based model with a stretched grid. Since re-analysis data at about 2- 5 km spatial resolution is available for parts of the Earth, it would indeed be possible to combine these with data from ERA5 at 0.25° resolution in a single model. Work in this direction is currently in progress, but to our knowledge no results have yet been shown. At this stage, it is therefore difficult to draw any conclusions both on deterministic and probabilistic/ensemble initial-state approaches trained on high-resolution data. It could be that the relative merit of such ML models varies by parameter.

The role or scope of post-processing methods is also not obvious, in particular if there are no additional reference/target data available.

The following is added to the conclusions: "*Finally, there is ongoing work on training global graph-based machine learning models with kilometre scale resolution over regions of special interest and coarser resolution elsewhere. These models make use of long archives of high-resolution limited area reanalysis data in combination with global reanalyses like ERA5. Comparing these with limited area NWP models like MEPS will be of great interest.*"

**Further revisions**

The following has been changed or added
- Author contributions
- Links to computer codes and data
- Use of NPG Copernicus latex style/template